# Study on Electromagnetic Performance of La_0.5_Sr_0.5_CoO_3_/Al_2_O_3_ Ceramic with Metal Periodic Structure at X-Band

**DOI:** 10.3390/ma15228147

**Published:** 2022-11-17

**Authors:** Zhaoning Yang, Lu Gao, Wei Ren, Ruiduan Zhang, Yangyang Chen, Qian Zhou, Kai Sun, Ziqi Jie, Yanmin Jia

**Affiliations:** 1School of Science, Xi’an University of Posts and Telecommunications, Xi’an 710121, China; 2Department of Materials Research, AVIC Manufacturing Technology Institute, Beijing 100024, China; 3Shaanxi Huaqin Technology Co., Ltd., Xi’an 710121, China; 4School of Materials and Chemical Engineering, Xi’an Technological University, Xi’an 710021, China

**Keywords:** periodic structure, La_0.5_Sr_0.5_CoO_3_/Al_2_O_3_ material, electrical performance, reflectivity

## Abstract

A radar absorbing material (RAM) is designed by combining the La_0.5_Sr_0.5_CoO_3_/Al_2_O_3_ ceramic and the metal periodic structure. The phase constitution and the microscopic morphology of the La_0.5_Sr_0.5_CoO_3_/Al_2_O_3_ ceramic are examined, respectively. The electrical properties and magnetic properties of the La_0.5_Sr_0.5_CoO_3_/Al_2_O_3_ ceramic are also measured at the temperature range of 25~500 °C. Based on the experimental and simulation results, the changes in the reflection loss along with the structure parameters of RAM are analyzed at 500 °C. The analytical results show that the absorption property of the RAM increases with the increase in the temperature. When the thickness of the La_0.5_Sr_0.5_CoO_3_/Al_2_O_3_ ceramic is 1.5 mm, a reflection loss <−10 dB can be obtained in the frequency range from approximately 8.2 to 16 GHz. More than 90% microwave energy can be consumed in the RAM, which may be applied in the high temperature environment.

## 1. Introduction

With the development of radar detection technology, the survival ability of aircraft or other weapons are facing a great threat in war environments. In order to improve the survivability of weapons, radar absorbing material (RAM) technology as a means to effectively reduce the radar cross section (RCS) of military targets has been highly valued by military powers in the world [1,2,3]. Generally, the RAM must have the characteristics of being “thin”, “wide”, “light” and “strong” [4,5,6,7,8]. In addition, restricted by the working environment of the aviation weapon components (such as the engine tail nozzle, wing front of fighter, missile and other weapons and equipment, as well as its working environment often being a high temperature), there is a needed to develop a new RAM which can adapt to a high temperature environment. For traditional RAM, their absorbing properties mainly depend on the content of the absorbent [9,10,11]. Currently, the absorbents of magnetic materials include metallic iron powder, ferrite, hydroxy iron, iron wire fiber and other magnetic materials, and the absorption of electromagnetic wave mainly depends on the magnetic polarization effects such as the hysteresis loss, eddy current loss, domain wall resonance, aftereffect loss and residual loss [12,13,14,15]. The absorbents of dielectric materials include graphite powder, carbon nanotube, carbon fiber, calcium titanate, magnesium titanate, barium titanate, spinel, zinc oxide and other dielectric materials, and the dielectric loss is mainly caused by electric conductivity, ion transition, ion vibration and ion deformation, etc. [16,17,18,19,20]. In addition, the RAM is composed of the absorbing material and metal periodic structure surface, which are highly valued by many researchers due to their unique electromagnetic resonance coupling characteristics. The results show that the reflection and transmission performance of electromagnetic wave can be effectively regulated through changing the shape, size and period of the periodic structure [21,22,23]. At present, most of the research on the properties of the RAM are focused on the room temperature. Research on the electromagnetic coupling of the FSS at a high temperature is scarce. It makes the absorption performance of the RAM difficult to control and affect the research progress of the RAM [24,25,26,27].

In this work, the RAM consists of La_0.5_Sr_0.5_CoO_3_/Al_2_O_3_ materials and a metal periodic structure are designed and studied. La_0.5_Sr_0.5_CoO_3_, with the perovskite structure, is widely used in the ferroelectric memory and battery materials because of its electron and ion conductivity, and it can also be used as an absorbent as a wave absorbing material. The effects of the structural parameters of the metal periodic structure on the absorption performance of La_0.5_Sr_0.5_CoO_3_/Al_2_O_3_ materials are studied at a high temperature environment, and the electromagnetic coupling and energy loss density of the RAM are analyzed in the 25~500 °C temperature range.

## 2. Design and Experiment

Figure 1 exhibits the structural diagram of the RAM. The RAM is divided into three parts. The upper part is the periodic structure layer which is composed of periodic structure metal patches, the middle part is the magnetic ceramic layer, which is synthesized through using aluminum oxide (Al_2_O_3_, purity 99.9%) and lanthanum strontium cobaltite (LaSrCoO_3_, purity 99.9%) material with the mass ratio of 1:1. The magnetic ceramic used is provided by Shaanxi Huaqin Technology Co., Ltd., Xi’an, China. The lower part is the metal plate. The metallic part and the ceramic layer can be connected through plasma spraying technology. As shown in Figure 1, the radius of the circular patches is *a*. The period length of the circular patches is *C*. The thickness of the magnetic ceramic is *t*. The thickness of the periodic structure layer is *t*_0_. The RAM are designed and simulated through using the commercial electromagnetism simulating software (ANSYS HFSS 15.0). Based on the impedance matching, the design of the RAM models consists of two steps: designing the La_0.5_Sr_0.5_CoO_3_/Al_2_O_3_ model followed by designing the periodic structure metal patches. Finally, solving the parameter setting is presented through using HFSS [28,29,30]. The initial parameter dimensions of the absorber are *C* = 10 mm, *a* = 2 mm, *t* = 2 mm and *t*_0_ = 0.01 mm.

The phases of the magnetic ceramic layer were measured through employing an X-ray diffractometer (XRD, D/MAX2500 diffractometer, Rigaku, Japan). The surface morphology of the magnetic ceramic was characterized through using a scanning electron microscope (SEM, Model JSM-6360, Japan Electronics) [31,32]. The complex permittivity and the complex permeability of the magnetic ceramic were examined through using a vector network analyzer (Agilent Technologies E8362B PNA, Santa Clara, CA, USA).

## 3. Results and Discussion

Figure 2a,b exhibits the XRD patterns and the SEM photographs of the magnetic ceramic layer, respectively. In Figure 2a, it can be observed that the magnetic ceramic layer contains two phases of Al_2_O_3_ and La_0.5_Sr_0.5_CoO_3_. In Figure 2b, it is found that the magnetic ceramic has a high density and a few pores.

Figure 3 shows the measured electromagnetic parameters of the magnetic ceramic in the X-band under the temperature of 25~500 °C. Figure 3a–d are the real part and the imaginary part of the complex permittivity and complex permeability, respectively. It is observed that both the complex permittivity and the complex permeability of the ceramic increase with the rise in the temperature. The permittivity and the permeability will directly affect the electromagnetism wave loss of the RAM. The bigger the permittivity and the permeability, the greater the loss [33,34,35]. Thus, the changes in the complex permittivity and the complex permeability may affect the electromagnetic absorption properties of the La_0.5_Sr_0.5_CoO_3_/Al_2_O_3_ ceramic.

Figure 4 shows the simulated reflection loss (RL) of the La_0.5_Sr_0.5_CoO_3_/Al_2_O_3_ ceramic under temperature of 25~500 °C. The RL is an effective evaluation standard of the microwave absorbance capacity of the metal backed slabs of the material, and a low RL corresponds to a high absorption. In Figure 4, it is observed that the RL of the La_0.5_Sr_0.5_CoO_3_/Al_2_O_3_ ceramic decreases with the increase in the temperature. Thus, the absorption performance of the La_0.5_Sr_0.5_CoO_3_/Al_2_O_3_ ceramic increases with the increase in the temperature, the enhancement of the absorption is attributed to the increase in the complex permittivity and the complex permeability.

Figure 5 exhibits the RL of the RAM versus the frequency for a different period length *C* at 500 °C when the other parameters remain unchanged. In Figure 5, an absorption peak is observed between the frequency of 8.2 GHz and 12.4 GHz. The position of the absorption peak moves towards a high frequency. The minimum value of the reflectivity increases as the period *C* increases. When the period length *C* is equal to 10 mm, the reflection loss < −10 dB is obtained between the 8.5 GHz and 10.4 GHz. The minimum value of the reflection loss is −15 dB. Therefore, the optimal period length may be about 10 mm.

Figure 6 exhibits the effects of radius *a* on the RL of the RAM when the other structure parameters remain unchanged. From Figure 6, a resonance peak appears between 8.2 GHz and 12.4 GHz with a variable radius. As the radius increases, the minimum value of the absorption peak of the RAM first decreases and then increases. When the radius *a* is equal to 2 mm, the minimum value −16 dB of the reflection loss of the RAM and the largest absorption bandwidth < −10 dB of the reflection loss are obtained. Therefore, the optimal radius of the periodic structure may be about 2 mm.

Figure 7 shows the effects of the thickness *t* on the absorption properties of the RAM when the other parameters remain unchanged. As the thickness of the magnetic ceramic increases, the resonance absorption peak increases first and then decreases. When the thickness *t* increases, the frequency of the absorption peak moves to a low-frequency region and the absorption bandwidth of the reflection loss < −5 dB also increases. When the thickness of the magnetic ceramic is equal to 1.6 mm, the absorption property of the RAM is the best and the minimum value of the reflectivity of the RAM is −19 dB at ~12.5 GHz. Therefore, the optimal thickness may be about 1.6 mm.

In order to make the absorbing performance of the RAM the best, the structural parameters of the RAM are optimized in this work. The genetic algorithm is employed to solve the optimization problem. Figure 8 shows the optimized results of the RAM between the temperature range from 25 °C to 500 °C. From Figure 8, it is observed that the reflection loss < −5 dB is obtained between 12.4GHz and 18 GHz. As the temperature increases, the reflectivity of the RAM decreases. When the temperature is 500 °C, the reflection loss < −10 dB can be obtained in the X-band. The reflection loss curve of the RAM shows a double absorption peak structure in 8.2~18 GHz, and the frequencies of the absorption peak are located at 9 GHz and 13.2 GHz, respectively. The minimum value of the reflection loss is −39 dB at 13.2 GHz. Thus, a broad bandwidth of the reflection loss < −10 dB can be obtained in the frequency range from approximately 8 to 16 GHz. More than 90% microwave energy can be consumed in the RAM which can fill the requirement of a practical application. The optimal structural parameters are as follows: period *C* is 10.31 mm, thickness *t* is 1.50 mm and size *a* is 1.59 mm.

Figure 9 shows the power loss density distribution of the RAM under different temperatures. In Figure 9, it is observed that the power loss density of the RAM increases as the temperature increases from 25 °C to 500 °C. The power loss density is directly related to the absorption efficiency of the RAM; the bigger the loss, the higher the absorption efficiency [36,37]. In Figure 9a–c, at the temperature range of 25~200 °C, the loss is mainly caused by the metal periodic structure layer. At the temperature range of 300~500 °C, the complex permittivity and the complex permeability of the La_0.5_Sr_0.5_CoO_3_/Al_2_O_3_ ceramic increases gradually, indicating that the loss is induced by the electromagnetic coupling of the ceramic layer and the periodic structure, as shown in Figure 8d–f. Thus, this result is consistent with Figure 8. Two absorption peaks are very strong when the temperatures are 300 °C, 400 °C and 500 °C, one of which is caused by the periodic structure layer and the other comes from the ceramic layer, as show in Figure 7. So, the resonant coupling is realized using the periodic structure layer and the magnetic ceramic for a wide-band absorption.

## 4. Conclusions

In summary, a new ultra-thin and wide-band radar absorbing material is designed and investigated. The temperature and frequency corresponding laws of the complex permittivity and the complex permeability of the magnetic ceramic are analyzed. The structure parameters of the metal periodic structure and ceramic thickness are studied. The results indicate that the complex permittivity and the complex permeability of the magnetic ceramic gradually increase, and the reflectivity of the absorber decreases with the increase in the temperature. The improvement of the electromagnetic absorption performance comes from the continuous strengthening of the resonance coupling in the metal periodic structure and magnetic ceramic.

## Figures and Tables

**Figure 1 materials-15-08147-f001:**
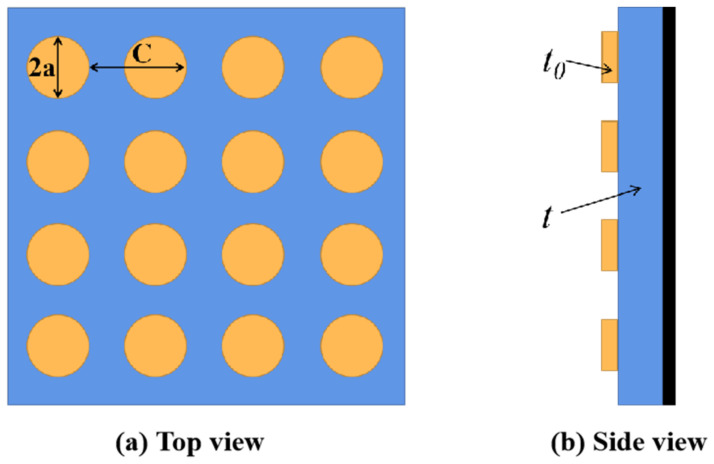
Structure diagram of RAM. The circular radius is *a* and the period length is *C*, the thickness of ceramic layer is *t*. (**a**) Top view, (**b**) side view.

**Figure 2 materials-15-08147-f002:**
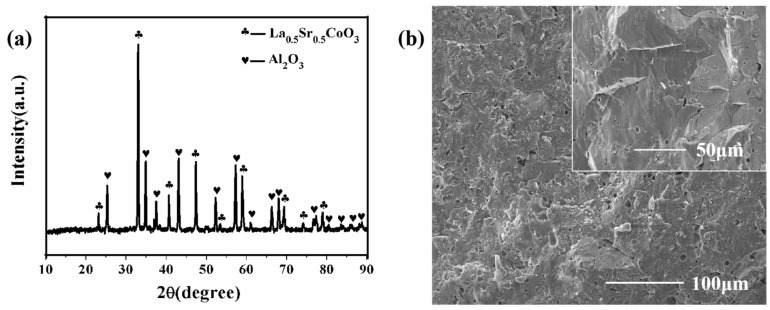
(**a**) X-ray diffraction patterns of magnetic ceramic. (**b**) Surface topography of magnetic ceramic.

**Figure 3 materials-15-08147-f003:**
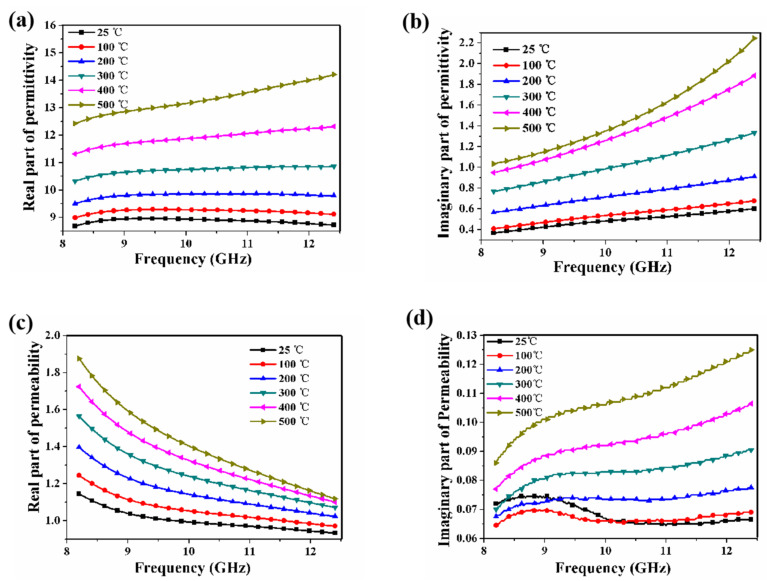
(**a**) Real part and (**b**) imaginary part of the complex permittivity of La_0.5_Sr_0.5_CoO_3_/Al_2_O_3_ ceramic. (**c**) Real part and (**d**) imaginary part of the complex permeability of ceramic layer.

**Figure 4 materials-15-08147-f004:**
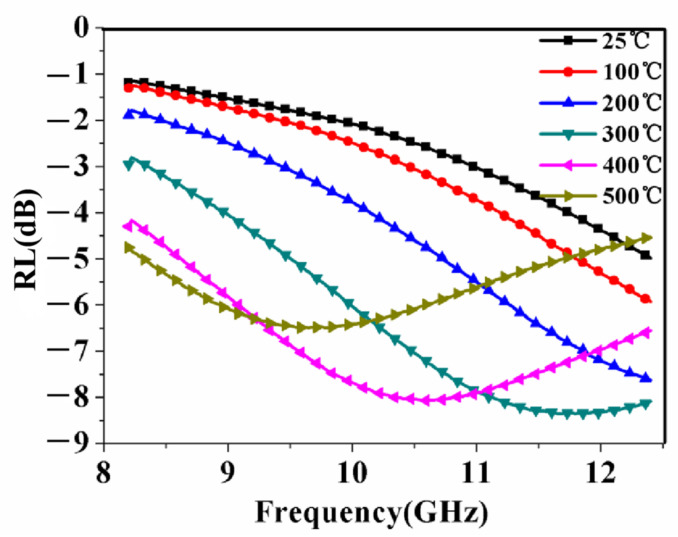
Reflection loss of La_0.5_Sr_0.5_CoO_3_/Al_2_O_3_ ceramic versus frequency with different temperature, *t* = 2 mm.

**Figure 5 materials-15-08147-f005:**
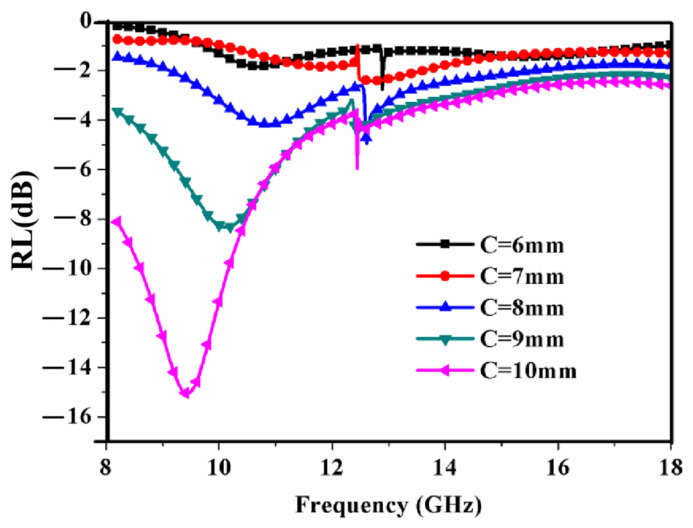
Simulated reflection loss of the RAM versus frequency for different *C*, *a* = 2 mm, *t* = 2 mm.

**Figure 6 materials-15-08147-f006:**
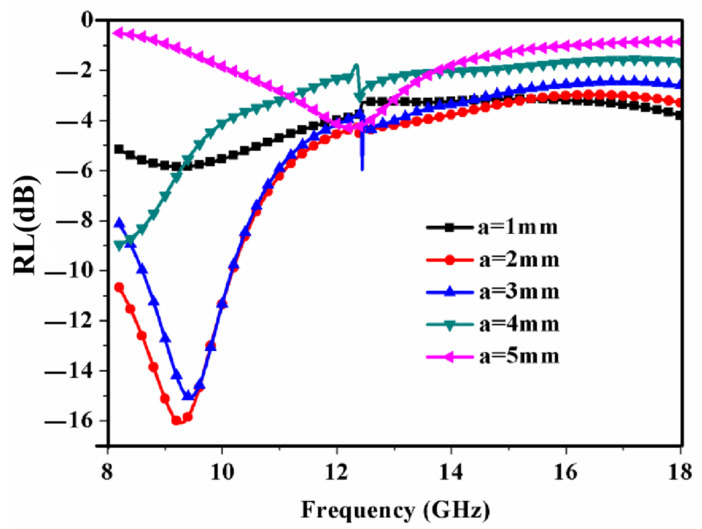
Simulated reflection loss of the RAM versus frequency for different *a*. *C* = 10 mm, *t* = 2 mm.

**Figure 7 materials-15-08147-f007:**
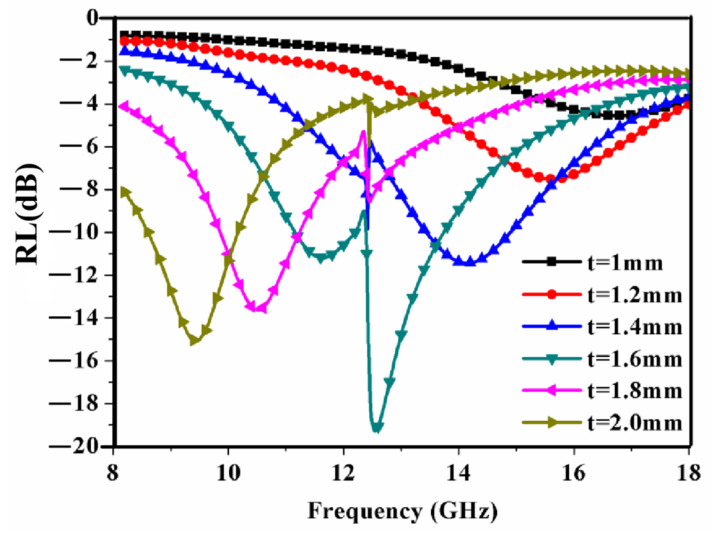
Simulated reflection loss of the RAM versus frequency for different *t*. *C* = 10 mm, *a* = 2 mm.

**Figure 8 materials-15-08147-f008:**
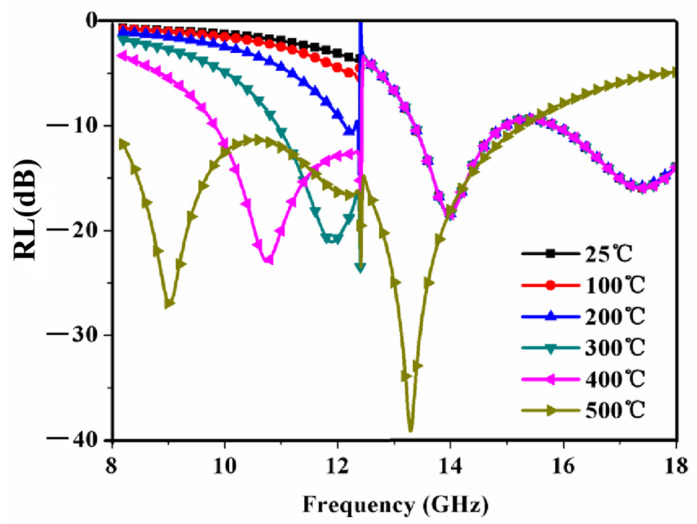
Optimized reflection loss results of RAM versus frequency for different temperature.

**Figure 9 materials-15-08147-f009:**
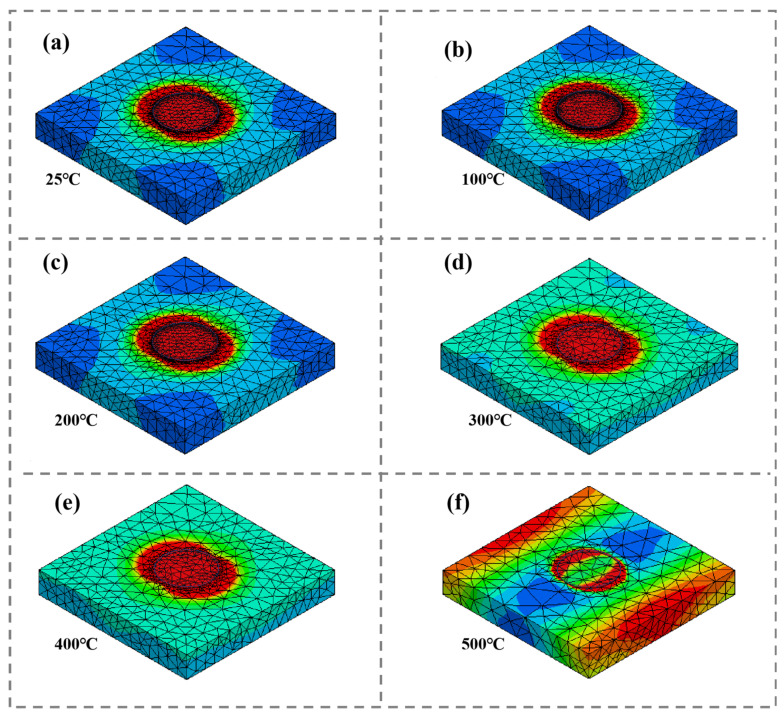
The volume power loss density distribution of RAM at different temperatures. (**a**) 25 °C, (**b**) 100 °C, (**c**) 200 °C, (**d**) 300 °C, (**e**) 400 °C and (**f**) 500 °C.

## Data Availability

Not applicable.

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
