# Peer review of "Study on Electromagnetic Performance of La0.5Sr0.5CoO3/Al2O3 Ceramic with Metal Periodic Structure at X-Band"

_materials, 2022, doi:10.3390/ma15228147_

Round 1

Reviewer 1 Report

This paper has studied a radar-absorbing material (RAM) designed by combining the La0.5Sr0.5CoO3/Al2O3 ceramic and the metal's periodic structure. Before publication, the manuscript needs a minor revision in the light of the following aspects:

1. The introduction, discussion, and conclusion must be improved by adding more background information to the introduction and providing more compression of the achieved results with previous studies.

2. Along the same lines, a graphical abstract may also help to draw the readers' attention with a specific focus of the review.

Reviewer 2 Report

This study, a radar absorbing material (RAM) consisting of La0.5Sr0.5CoO3/Al2O3 materials and metal periodic structure are designed and studied. This material can also be used as an absorbent as a wave absorbing material. Moreover, the effects of structural parameters of metal periodic structure on the absorption performance of La0.5Sr0.5CoO3/Al2O3 materials are analyzed in the 25 ~ 500 ℃ temperature range. 

The manuscript is well organized, but the results should be more correlated. 

After a careful reading of the manuscript, there are some observations and corrections to be made. 

1) In introduction, please explain more in details (special work conditions) why is necessary to study the behavior of the absorbent material at high temperature. 

2) Regarding the design process of the RAM material, the steps are not clear mentioned, how is realized the connection between the metallic part and the ceramic layer. Which is the nature of the metal used for the lower and upper parts? I think that is an important aspect. 

3) Fig. 1 exhibit the structural diagram of RAM – please replace with “exhibits” 

4) “In Fig. 2 (b), it’s found that the magnetic ceramic has a high density and a few pores” How important is the presence of a certain porosity for the RAM material?

The subject is very actual, in the present situation all over the world, maybe it should be more poited out the relevance of this kind of materials.

Round 2

Reviewer 2 Report

The manuscript was improved after the revision, according to the reviewer suggestion. Now it is more concludent.